# Associations between Disease Activity/Severity and Damage and Health-Related Quality of Life in Adult Patients with Localized Scleroderma—A Comparison of LoSCAT and Visual Analogue Scales

**DOI:** 10.3390/jcm9030756

**Published:** 2020-03-11

**Authors:** Anna Lis-Święty, Alina Skrzypek-Salamon, Irmina Ranosz-Janicka, Ligia Brzezińska-Wcisło

**Affiliations:** Department of Dermatology, School of Medicine in Katowice, Medical University of Silesia, Francuska 20/24, 40-027 Katowice, Poland; skrzypekalina@interia.pl (A.S.-S.); irmina.janicka@gmail.com (I.R.-J.); kikderm@sum.edu.pl (L.B.-W.)

**Keywords:** localized scleroderma, morphea, outcome measure, skin score, quality of life

## Abstract

Localized scleroderma (LoS) is a chronic fibrosing disorder of the skin and, less commonly, subcutaneous tissues. As the disease causes subjective symptoms, cosmetic defects, and, at times, functional disability, subjects with LoS experience deterioration of their health-related quality of life (HRQoL). The influence of disease activity/severity and damage status on HRQoL measures in patients with LoS is scarcely known. Physician-reported measures (modified LoS skin severity index, LoS skin damage index, physician global assessments of the disease activity/severity and damage) and patient-derived measures (patient global assessments of the disease activity/severity and damage) were obtained in adult LoS patients. Their HRQoL was measured with Skindex-29 and Short Form-36. The patients’ assessments of disease activity/severity and damage in LoS differed from the assessments by the physicians. The patients’ predominant concerns centered on LoS-related damage, whereas the physicians’ concerns focused on features of disease activity. Visual analogue scales bore some relation to the HRQoL, and they seem to be important in a holistic approach to the patient and should not be omitted in LoS evaluation.

## 1. Introduction

Localized scleroderma (LoS) is an autoimmune connective tissue disorder characterized by inflammation and fibrosis of the skin and, less commonly, deeper tissues (subcutaneous fat, fascia, muscles, and occasionally even bones) in the affected area. This is a rare disease, with two peak incidence periods—between the ages of 7 and 10 and between 40 and 50 with female preponderance (3:1–4:1) [1]. LoS has a wide variety of clinical presentations in both age groups. Limited, generalized, linear, deep, and mixed forms can be diagnosed [2]. At the beginning of the disease, the skin lesions are usually inflamed plaques that become progressively indurated (activity status). They may be accompanied by subjective sensations (e.g., pruritus, pain) and stay active for several months to years, then resolving with atrophy of the skin and subcutaneous tissues, leaving hyper/hypopigmentation (damage status). Additionally, in the early phase of LoS, arthralgia and/or arthritis (mostly related to the site of the skin lesions) may be observed, especially in the linear forms of the disease [3,4]. Subsequently, joint contractures of the affected limb, limb length discrepancy, and physical disability may occur, mainly in children [5]. Neurological symptoms (e.g., seizures/epilepsy, headaches/migraines, stroke) and/or eye conditions (such as uveitis, xerophthalmia, papilledema, episcleritis, glaucoma, cataract) have been reported in some LoS cases in the head or face [4,5]. The frequency of these manifestations differs depending on age at disease onset and is highest in childhood-onset LoS [1,4,5]. Other autoimmune disorders may coexist with LoS, e.g., Hashimoto’s disease, Grave’s disease, rheumatoid arthritis, or lichen sclerosus, mainly in adult patients [1,6]. Our systematic review showed that this group of patients had a higher number of poor health-related quality of life (HRQoL) scores than juvenile patients, the latter often having a less favorable course of LoS [7]. Factors considered by physicians and patients when evaluating physical health differ. HRQoL may be affected by disease activity/severity and damage status, but their influence from the perspective of patients and physicians is scarcely known.

The localized scleroderma cutaneous assessment tool (LoSCAT) is a clinician-reported measure that considers the cutaneous features of the disease activity/severity and damage [3,7,8]. However, the cutaneous involvement does not seem to reflect the general severity of LoS. It is worth emphasizing that a number of studies detail the huge impact of the extracutaneous manifestations, subjective symptoms, and comorbidities on the HRQoL of LoS patients [9,10,11,12,13,14]. Visual analogue scales (VASs) are the only available tools that seem to catch all aforementioned issues. For LoS, VASs were firstly used to assess itchiness and tightness [9], and then by Arkachaisri et al. [3,8] in a global assessment of juvenile LoS. The authors measured the physician (Phys) and patient (Pt) global assessment (GA) of disease activity (A)/severity (S) and damage (D) over a one-month period [3,8]. 

The objective of our study was to compare the relationship of LoSCAT and VASs to HRQoL in adult patients.

## 2. Experimental Section 

### 2.1. Materials and Methods 

#### 2.1.1. Study Design and Patient Selection 

This cross-sectional study was approved by the Medical University of Silesia (ref. KNW/0022/KB1/134/15). Study participants were recruited from the dermatology department at the Medical University of Silesia in Katowice. All subjects gave written informed consent before their inclusion in the study. The patients were diagnosed with LoS according to the classification of Kreuter et al. with a biopsy specimen when necessary [2]. Data were collected using a structured questionnaire for demographic and clinical information. Patients were asked about their age, age at the disease onset and at the LoS diagnosis, number of recurrences, extracutaneous manifestations (arthralgia, swollen joints, headache, decreased vision, any other neurological or eye problems), subjective symptoms (itchiness, pain, burning or pricking sensation), and coexisting disorders. A detailed physical examination was undertaken. All patients were evaluated by a consultant ophthalmologist and neurologist and those with arthralgia by a rheumatologist also. In all patients, a erythrocyte sedimentation rate (ESR) and serum C-reactive protein (CRP) level was determined at admission with routine methods. 

#### 2.1.2. LoSCAT and VASs

Cutaneous features of the activity of the disease and tissue damage were assessed with the modified LoS skin severity index (mLoSSI) and LoS skin damage index (LoSDI) as described previously [15]. The characteristics determining global disease activity and damage in adult LoS patients were identified to properly evaluate the PhysGA and PtGA of the disease activity/severity and damage [1,3,8]. To simplify and standardize PhysGA, the lack of activity and mild/moderate/high and very high activity as well as mild/moderate/high and very high degree of damage in LoS were defined (Table 1 and Table 2). Information about LoS and features determining disease activity/severity and damage was prepared for patients. A standard instruction on how to complete the PtGA-S and PtGA-D forms was mentioned. When patients filled out the PtGA-S, they answered the question: “How severe was your localized scleroderma during last month”. The VAS line was anchored by two verbal descriptors: “not severe” and “very severe”. Answers were made by making a vertical line on the 100 mm VAS that corresponded to the numerical score. When patients filled out the PtGA-D they answered the question: “How do you assess a damage of your skin caused by localized scleroderma during last month”. Patients were asked to place a line perpendicular to the VAS line at the point that represented their damage severity between 0 (“no damage”) and 100 (“very severe damage”). 

Two dermatology residents were trained in the correct use of the described measures (mLoSSI, LoSDI, PhysGA-A, and PhysGA-D) by an expert (A.L.S.). Both physicians examined each patient twice, firstly at admission and again two days later. Patients were asked to complete the PtGA-S and PtGA-D twice at similar time intervals. 

#### 2.1.3. HRQoL

Patients’ HRQoL was evaluated using the Short form-36 questionnaire (SF-36v2, license number QM038248) and the Polish version of the Skindex-29 (with the permission of Konrad Janowski, PhD). Both questionnaires are widely used to assess levels of HRQoL. From the SF-36v2 questionnaire, a physical component score (PCS) and a mental component score (MCS) can be calculated. In SF-36v2, there is no single overall score for the Skindex-29, but items are arranged into three subscales: emotions, functioning, and symptoms. All patients completed both questionnaires in regards to the previous four weeks. The impact of LoS on psychosocial functioning as measured by the Skindex-29 was taken into consideration for the PhysGA-D. The results of the Skindex-29 were interpreted as mild, moderate, or severe HRQoL impairment, following Prinsen et al. [16]. 

#### 2.1.4. Statistical Analysis

Descriptive statistics were expressed mainly as the median and interquartile range (and also as means and standard deviations for the clinical data) for the continuous variables, and as numbers and percentages for the categorical variables. Spearman’s correlation, Kruskall–Wallis, and Mann–Whitney U tests were used to analyze the relationships of the clinical parameters to HRQoL. The reliability and concurrent validity of the mLoSSI, LoSDI, PhysGA-A, PhysGA-D, PtGA-S, and PtGA-D were assessed and interpreted according to previously described characteristics [15]. 

Inter-rater reliability was determined with the use of Cohen’s kappa statistics and values were indicative of poor reliability (< 0), slight (0–0.2), fair (0.21–0.4), moderate (0.41–0.6), substantial (0.61–0.8), and almost perfect (0.81–1) [17]. Spearman’s correlation coefficient was used to assess intra-rater reliability and values were interpreted as follows: low positive (≥0.3, <0.5), moderate (≥0.5, <0.7), strong/high degree of agreement (≥0.7, <0.9), and very strong/excellent agreement (≥0.9) [17]. Convergent validity of the LoSCAT and relationship of its components and VASs to HRQoL measures were also determined with the use of Spearman’s rank correlation coefficient with similar cut-off values. The 95% confidence level was used.

All data were analyzed using Statistica 12.0 software (StatSoft, Inc., Tulsa, OK, USA) and PQStat (v.1.6.2; PQStat, Poznań, Poland) with *p* < 0.05 being statistically significant. 

## 3. Results

### 3.1. Study Participants Characteristics 

The research included 40 patients with LoS: 18 (45%) with plaque, 8 (20%) with atrophoderma Pasini et Pierini, 7 (17.5%) with generalized, 4 (10%) with mixed, and 3 (7.5%) with linear subtype (two patients with en coup de sabre and one patient with linear LoS involving the lower limb). The majority of patients (33 of 40, 82.5%) were females. All patients were adults at the time of survey (mean 49, range 19–81 years old). The mean age of the patients was 42.4 years at the disease onset, and most of them (38 of 40, 95%) had the disease onset in adulthood. Only two patients had the disease onset at the age of four and five years old, and after a long period of quiescence their LoS relapsed when they were at the age of 31 and 22 years old, respectively. The mean duration of the disease for the study group was 6.6 years, with a minimum of one month and a maximum of 30 years. The mean body surface area affected by LoS was 6.7%. Most patients (35 of 40, 87.5%) had the active disease (mLoSSI > 0), of which six patients were experiencing a first and three patients a second relapse, but the rest were in their first episode of the disease activity. One woman with mixed (coexistence of linear and plaque variants) and another one with a generalized subtype of LoS reported arthralgias, as did a man with linear LoS of the limbs who also had joint contractures. More than half of patients (21 of 40, 52.5%) reported symptoms, such as skin pain, itchiness, tightness, or burning sensations, in the affected area at least once during the last month. Other extracutaneous manifestations, such as neurological or ocular symptoms were absent. Twenty-five patients had comorbid medical conditions, most frequently: arterial hypertension (*n =* 9), diabetes type 2 (*n =* 5), gastroesophageal reflux (*n =* 3), Hashimoto’s disease (*n =* 2), rheumatoid arthritis (*n =* 2), and hand eczema (*n =* 2). CRP was mildly elevated in one patient with active mixed (6 mg/l) and one with active generalized (6.6 mg/L) LoS. ESR was abnormal in 10 patients (six had generalized, three had plaque and one had mixed subtypes) of which nine (90%) had active LoS. The clinical evaluation scores for the overall sample are presented in Table 3.

### 3.2. Reliability and Concurrent Validity of LoSCAT and VASs

The results confirmed a satisfactory reliability of LoSCAT, PhysGA-A, and PhysGA-D. Intra and inter-rater reliability of the mLoSSI and LoSDI were found to be excellent and were reported previously [15]. The inter-rater reliability of the PhysGA-A as well as PhysGA-D was substantial (Kappa = 0.74 and 0.71 respectively, *p <* 0.01). The intra-rater reliability of the PhysGA-A was higher (r_s_ = 0.89, *p <* 0.01) that that of the PhysGA-D (r_s_ = 0.71, *p <* 0.01), but both results showed a strong agreement between examined characteristics. A high degree of agreement between intra-observer scores for the PtGA-S (r_s_ = 0.87, *p <* 0.01) and PtGA-D (r_s_ = 0.89, *p <* 0.01) also provided proof of strong intra-observer reliability of both scales.

A satisfactory correlation between VASs and LoSCAT proved the adequate concurrent validity of the scales. The physician-derived activity measures correlated moderately with each other (PhysGA-A and mLoSSI: r_s_ = 0.64, *p <* 0.01), while a low positive correlation was found between PhysGA-D and LoSDI (r_s_ = 0.47, *p <* 0.01). A similar relationship was found between the physician- and patient-derived global assessments. There was a moderate correlation between the PhysGA-A and PtGA-S (r_s_ = 0.65, *p <* 0.01), and a low correlation between the PhysGA-D and PtGA-D (r_s_ = 0.42, *p* = 0.01). The patient-derived measures were less associated with the LoSCAT than the physicians. The correlation of the PtGA-S with mLoSSI was low (r_s_ = 0.46, *p <* 0.01), and between the PtGA-D and LoSDI it was very low (r_s_ = 0.24, *p >* 0.1). 

### 3.3. Relationship between Clinical Assessment and HRQoL

No significant associations were found between the mLoSSI and LoSDI or their domains and HRQoL at all. Higher scores for the PhysGA-A moderately correlated with worse HRQoL as measured by the SF-36:PCS (r_s_ = −0,44, *p <* 0.01) (Figure 1), but there was no correlation found with the PhysGA-A and SF-36:MCS nor Skindex-29. PtGA-D was the only measure weekly to moderately associated with all Skindex-29 subscales (emotion: r_s_ = 0.36, *p* = 0.01; functioning: r_s_ = 0.38, *p* = 0.05, and symptoms: r_s_ = 0.48, *p* = 0.05) (Figure 2) and the SF-36:MCS (r_s_ = −0,32, *p <* 0.05) (Figure 3), but not with SF-36:PCS. There were no relationships whatsoever between HRQoL and two of the GAs, namely the PhysGA-D and PtGA-S.

## 4. Discussion

In the current study, we assessed LoS activity/severity and damage using LoSCAT and VASs designed for physicians and adult patients. Then, we compared these scales with each other, and their relationships to HRQoL. The activity/severity and damage scored by physicians and patients differed. A moderate relationship was found between the global assessments of LoS activity/severity derived from physicians and patients (PhysGA-A and PtGA-S), but only a low correlation between mLoSSI and PtGA-S was shown. A low correlation was observed between physician and patient global assessments of LoS-related damage (PhysGA-D and PtGA-D) and a lack of relationship was found between LoSDI and PtGA-D. The two latter results indicated the possibility of physicians underestimating the damage in LoS. Similarly, mLoSSI appeared to not reflect the activity of the disease. The extensive clinical heterogeneity of LoS, presence of subjective symptoms, extracutaneos manifestations, lack of available specific laboratory tests, and psychoemotional impact of the disease hinder its appropriate assessment. Both indexes—the mLoSSI and LoSDI—are based on the physical examination of LoS skin lesions at 18 cutaneous anatomic sites and measure the exact intensity of the listed lesions features: new/enlarged lesion (within 1 month), erythema, skin thickness, dermal or subcutaneous atrophy, and dyspigmentation (hyper/hypopigmentation) [3,8,15]. Although the LoSCAT provides an objective and reliable assessment of skin lesions, the scale does not take into account the subjective symptoms (skin pain, itchiness, tightness, or burning sensations) and extracutaneous manifestations of the disease. Therefore, we designed VASs to assess the overall disease activity and damage in our adult patients with LoS. Analogously to the LOCUS group recommendations [3,8], we adapted VASs so they included not only factors related to cutaneous variables, but also to subjective symptoms, extracutaneous features, and laboratory test results. A presence of possible functional limitations, orthopedic, ocular, or neurologic symptoms and the impairment of HRQoL are all variables that were used to determine the global assessments of the damage in adult-onset LoS. As was seen in previous reports [3,8], our study showed satisfactory inter- and intra-observer agreement of the physician global assessments. The intra-observer agreement of the patient global assessments was also good. The reliability of VASs was generally inferior to that of the LoSCAT. The huge number of variables most likely hindered the determination of the disease in the physician global assessments and was a factor that reduced their reliability, but this consequently provided more information about LoS. 

As this study and earlier studies showed [15,18], mLoSSI and LoSDI does not correlate with HRQoL measured with the Skindex-29 and SF-36 questionnaires, or such correlation was poor in adult patients. Similarly, both indexes also did not significantly influence the HRQoL in juvenile LoS studies, when the children dermatology life quality index was assessed [3,8,11,13]. According to Klimas et al. [10], the mLoSSI and PhysGA-A were more closely linked with Skindex-29 and SF-36:PCS scores than the LoSDI and PhysGA-D in the Morphea in Adults and Children cohort. In accordance with the aforementioned results, a moderate correlation of the PhysGA-A with SF-36:PCS impairment was found in our patients. Several studies showed that pain, itchiness, and burning sensations were related with the impairment of the HRQoL in the PCS domain of SF-36 [9,10,11,12]. Such symptoms occurred in most patients participating in the present study (52.5%). Similarly to our observations, in studies where the dermatology life quality index scores were correlated with the LoSCAT and VASs, the results suggested that only PhysGA-A was linked with HRQoL [10,19]. There was a lack of significant correlation between PhysGA-D and HRQoL results in our study. It should be noted that despite the low incidence of limbs or face–head involvement, and associated extracutaneous manifestations, study participants had generally poor HRQoL scores and this was one of the factors taken into account in the PhysGA-D. In contrast, there was no relationship between the PtGA-S and HRQoL measured by the skin-specific as well as universal scale, and the PtGA-D correlated with all subscales of Skindex-29 and also with the SF-36:MCS. This suggests that HRQoL was related more to the patient’s evaluation of the disease damage than to the patient’s assessment of the disease activity/severity. 

The proper assessment of changes in the activity of LoS is a fundamental priority in clinical trials and daily practice. Our study reports, for the first time, the physician–patient discrepancies in their assessments of LoS. Damage related to LoS, as perceived by patients, shows how they think about their own condition and does not necessarily express objective health status. We would like to highlight that the VASs evaluate the disease from a wider perspective and bear some relation to the HRQoL of adult patients, in opposition to LoSCAT, which leaves out this subject. Therefore, the PhysGA-A seems to be more useful than the mLoSSI as a measurement of disease activity. It should be also added that the PtGA-D may initially give indirect information about the psychosocial functioning of adults with LoS. A limitation of this study was the lack of a juvenile LoS group and an extensive analysis of factors influencing HRQoL. Understanding the needs of patients with different characteristics may help to strengthen the patient–physician relationship and deserves further investigation. HRQoL is affected by physical health, emotional and social relationships, economic status, surrounding environment, and life satisfaction. Thus, future, larger investigations into all the factors should be pursued.

## 5. Conclusions

The physician’s and patient’s assessment of LoS activity/severity and damage were discrepant. HRQoL of adult patients may be affected more by LoS-related damage than the activity/severity status. The use of VASs may be useful for a comprehensive LoS evaluation in adulthood and may provide a holistic approach to the medical care of a patient.

## Figures and Tables

**Figure 1 jcm-09-00756-f001:**
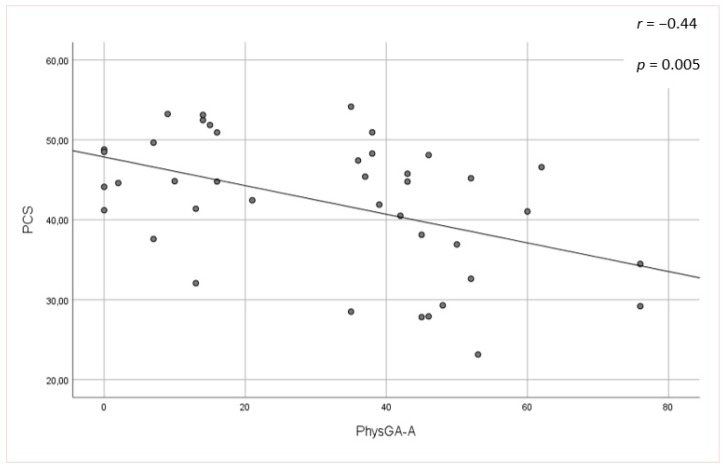
****** Relationship between physician global assessment of the disease activity (PhysGA-A) and physical component summary (PCS) of the Short Form-36 in localized scleroderma patients. *p*-value, r—Spearman’s rank correlation coefficient.

**Figure 2 jcm-09-00756-f002:**
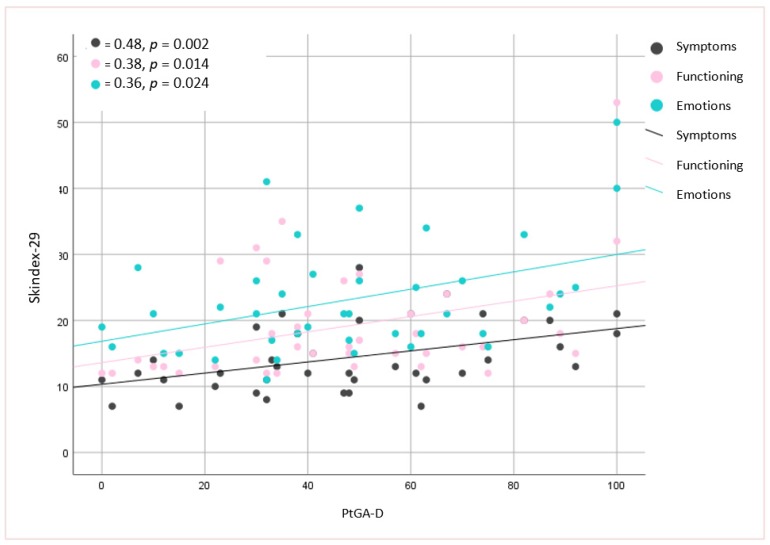
****** Relationship between patient global assessment of the disease damage (PtGA-D) and Skindex-29 in localized scleroderma patients. *p*-value, r—Spearman’s rank correlation coefficient.

**Figure 3 jcm-09-00756-f003:**
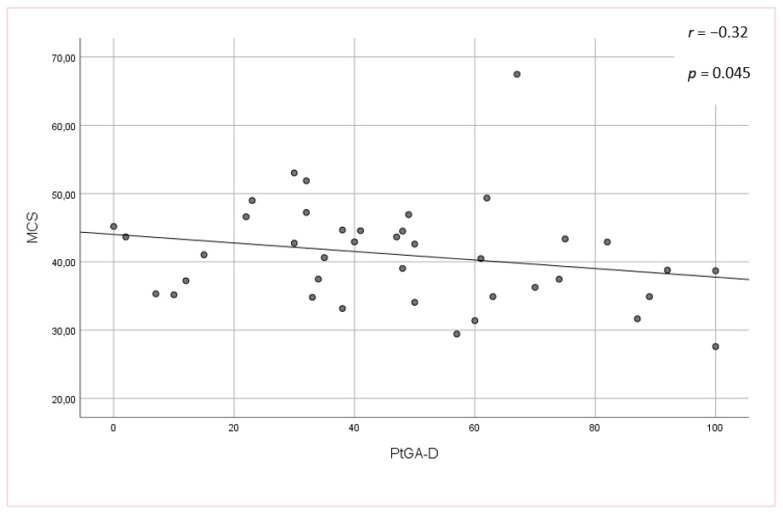
Relationship between patient global assessment of the disease damage (PtGA-D) and mental component summary (MCS) of the Short Form-36 in localized scleroderma patients. *p*-value, r—Spearman’s rank correlation coefficient.

**Table 1 jcm-09-00756-t001:** Features determining physician global assessment of the activity (PhysGA-A) of localized scleroderma.

Variable	Level of Localized Scleroderma Activity				
	Lack	Mild	Moderate	High	Very High
New or enlarged skin lesions	-	-	+	+	+
Erythema	-	+	+	+	+
Skin thickness	-	+/−	+/−	+/−	+/−
Subjective symptoms (e.g., itch, pain, burning sensations)	-	-	-	+/−	+/−
General symptoms (e.g., weakness, arthralgia, uveitis) *	-	-	-	+	+
Laboratory signs (e.g., elevated erythrocyte sedimentation rate and/or C-reactive protein level) *	-	-	-	-	+

* When caused most probably by localized scleroderma, not other underlying disease.

**Table 2 jcm-09-00756-t002:** Features determining physician global assessment of the damage (PhysGA-D) in localized scleroderma.

Variable	Level of Damage in Localized Scleroderma				
	Lack	Mild	Moderate	High	Very High
Dyspigmentation	-	+	+	+	+
Dermal atrophy	-	-	+	+	+
Subcutaneous atrophy	-	-	+/−	+	+
Muscle atrophy	-	-	-	+/−	+/−
Bone atrophy	-	-	-	+/−	+/−
Functional limitations due to extracutaneous manifestations (e.g., orthopedic, neurological, eye complications)	-	-	-	-	+
Decreased HRQoL *					

* HRQoL—Health-related quality of life assessed by Skindex-29.

**Table 3 jcm-09-00756-t003:** Clinical characteristics of 40 patients with localized scleroderma.

Item	Min-Max	Mean (±SD)
mLoSSI	0–29	7.15 (±7.21)
New or enlarged skin lesions (N/E)	0–6	0.83 (±1.66)
Erythema (ER)	0–19	3.48 (±3.78)
Skin thickness (ST)	0–15	2.85 (±3.534)
LoSDI	0–52	10.42 (±9.82)
Dermal atrophy (DAT)	0–25	4.28 (±4.64)
Subcutaneous atrophy (SAT)	0–12	1.55 (±2.76)
Dyspigmentation (DP)	0–15	4.6 (±3.59)
PhysGA-A	0–76	31.35 (±21.58)
PhysGA-D	0–88	39.20 (±15.42)
PtGA-S	0–100	42.48 (±31.42)
PtGA-D	0–100	47.63 (±26.83)

mLoSSI—modified Localized Scleroderma Skin Severity Index, LoSDI—Localized Scleroderma Skin Damage Index, PhysGA-A—Physician Global Assessment of Disease Activity, PhysGA-D—Physician Global Assessment of Disease Damage, PtGA-S—Patient Global Assessment of Disease Severity, PtGA-D—Patient Global Assessment of Disease Damage.

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
