# Peer review of "Associations between Disease Activity/Severity and Damage and Health-Related Quality of Life in Adult Patients with Localized Scleroderma—A Comparison of LoSCAT and Visual Analogue Scales"

_jcm, 2020, doi:10.3390/jcm9030756_

Round 1
Reviewer 1 Report
This is a well-designed cross-sectional study that compares LoSCAT and visual analogue scales and investigates the influence of disease activity/severity and damage on health-related quality of life in localized scleroderma.
Minor comments:
"visual analogue scales" instead of "visual analoque scales" please add and briefly comment the following reference in the Introduction section --> Marzano AV et al. Eur J Dermatol; 13:171-6Author Response
Dear Reviewer,
Thank you for your important advices. The suggested changes have taken place in the introduction part of the manuscript (lines 41-45). The correction of typographical errors has been made.
Sincerely,
Anna Lis-Święty
Reviewer 2 Report
In this study, the authors investigated various kinds of LoS-associated clinical indexes. The main message of this study is the physician-patient discrepancies in their assessment of LoS. This point is clearly demonstrated in this study. The manuscript is well conceived, well written and contains interesting results. This reviewer has just a minor comment before publishing this paper.
1. The patient population enrolled in this study is different from the general population of LoS. According to previous studies, the proportion of juvenile LoS patients is much higher than that reported in this study. This point should be described as a limitation of this study.
Author Response
Dear Reviewer,
Thank you for your valuable suggestions. I have included a sentence on the limitation of the study at the end of the manuscript (lines 265-266). I have changed the title of the manuscript and less precise sentences to show that they are related to adult population (lines 209, 274).
Sincerely,
Anna Lis-Święty
Reviewer 3 Report
This is very well written and very important paper with clear observations.
I have some minor remarks:
Was the joint involvement assessed by a rheumatologist?
Was an uveitis screening program installed?
Table 1. and 2. – could the extracutaneous involvement added?- it would be important to have this
Extracutaneous manifestations were assessed, how was the correlation of that ot QOL?
Subjective symptoms were assessed, like itching.. . How is the correlation of this to QOL measures?
Author Response
Dear Reviewer,
Thank you for your helpful comments. I have provided examples of extracutaneous involvement in Table 1. and 2. All patients were evaluated by consultant ophthalmologist and neurologist and those with arthralgia also by a rheumatologist (lines 75-76). Multiple-factor logistic regression analysis confirmed, as with single-factor analysis, that subjective symptoms (pruritus, pain, paresthesia) and musculoskeletal manifestations were the causes of the low SF-36 physical component score. I’m going to describe these results in the article „Health-related quality of life and its influencing factors in adult patients with localized scleroderma”. Up to the present article, I have added critical summary at the end of the manuscript and I have highlighted the possible directions for future research (lines 265-270). I have changed the title of the manuscript and less precise sentences to show that they are related to adult population (lines 209, 274).
Sincerely,
Anna Lis-Święty